# Tunable Composition of Dynamic Non-Viral Vectors over the DNA Polyplex Formation and Nucleic Acid Transfection

**DOI:** 10.3390/polym11081313

**Published:** 2019-08-06

**Authors:** Lilia Clima, Bogdan Florin Craciun, Gabriela Gavril, Mariana Pinteala

**Affiliations:** “Petru Poni” Institute of Macromolecular Chemistry, Romanian Academy, Centre of Advanced Research in Bionanoconjugates and Biopolymers, Grigore Ghica Voda Alley, 41 A, 700487 Iasi, Romania

**Keywords:** polyethyleneglycol, polyethyleneimine, pegylated squalene, dynamic combinatorial chemistry, pDNA condensation, non-viral vectors, gene therapy, DNA transfection

## Abstract

Polyethylene glycol (PEG) functionalization of non-viral vectors represents a powerful tool through the formation of an overall surface charge shielding ability, which is fundamental for efficient nucleic acid delivery systems. The degree of non-viral vector PEGylation and the molecular weight of utilized PEG is crucial since the excessive use of PEG units may lead to a considerable reduction of the DNA-binding capacity and, subsequently, in a reduction of in vitro transfection efficiency. Herein, we report a detailed study on a series of dynamic combinatorial frameworks (DCFs) containing PEGylated squalene, poly-(ethyleneglycol)-bis(3-aminopropyl) of different lengths, and branched low molecular weight polyethylenimine components, reversibly connected in hyperbranched structures, as efficient dynamic non-viral vectors. The obtained frameworks were capable of forming distinct supramolecular amphiphilic architectures, shown by transmission electron microscopy (TEM) and dynamic light scattering (DLS), with sizes and stability depending on the length of PEG units. The interaction of PEGylated DCFs with nucleic acids was investigated by agarose gel retardation assay and atomic force microscopy (AFM), while their transfection efficiency (using pCS2+MT-Luc DNA as a reporter gene) and cytotoxicity were evaluated in HeLa cells. In addition, the data on the influence of the poly-(ethyleneglycol)-bis(3-aminopropyl) length in composition of designed frameworks over transfection efficiency and tolerance in human cells were analyzed and compared.

## 1. Introduction

Highly effective DNA delivery systems are multivalent “nanomachines”, with active functionalities integrated and positioned at the nanoscale [1]. Until present, various functional platforms have already been rationally designed and synthesized with the hope of mimicking the complicated biological mechanisms [2,3]. However, DNA and targeted cells are highly variable systems and rational design is limited to a relatively small number of components. One possible solution to this problem is to employ the constitutional dynamic chemistry (CDC) as a new evolutional approach to produce chemical diversity [4,5]. In recent years, our group has been working on developing and implementing particular strategies to create a diversity of modular vectors capable of dynamically self-adapting to their DNA targets, allowing for the rapid screening of most effective vectors, optimally matched to DNA 3D surrounding space [6,7,8,9,10]. The use of reversible interactions as dynamic interfaces between the target and dynamical constitutional frameworks (DCF) components strongly contributed to the self-adjustment of the system’s tridimensional geometry and functional properties. Thus, a number of adaptive dynamic vectors based on polyethylene glycol, cationic moiety components and, in some cases, squalene derivative, reversibly connected to core centers were designed, prepared and tested as efficient non-viral vectors for DNA transfection [7,8,11]. Recently, we demonstrated the synergistic effect of low molecular weight polyethylenimine (PEI) and polyethylene glycol (PEG) components of a dynamic non-viral vector structure for a deeper understanding of the structure–performance relationship (in terms of transfection efficiency and tolerance in humans cells) in such systems [9]. The results have clearly shown the importance of PEG units’ presence in dynamic non-viral vectors formation, revealing a cooperative effect between PEI and PEG moieties. An important aspect regarding the presence of PEG in the structure of non-viral vectors, a subject that has intensively been studied in recent years, was the surface charge shielding effect of the vectors by this hydrophilic polymer [12]. PEG units of different molecular weights have been tested in the building of non-viral vectors and, subsequently, have been tested both in vitro and in vivo [13,14,15,16,17]. The main objective of the hydrophilic coating is to diminish the interactions of the particles with blood components and to reduce the uptake by macrophages and this, in turn, should result in an increased blood circulation time [18]. Various strategies have already been implemented in the design of non-viral vectors in order to find the optimum PEG amount per vector, ranging from variation of PEG units’ molecular weight [17] to the exploitation of specific hydrophobic interactions of attached ligands that are known to enhance the physical attraction between the components [19,20]. These adjustment strategies belong to what has been called “stealth technology”, and multiple results indeed showed that PEGylation of PEIs led to an increased solubility of the complexes as well as to an overall reduction of the surface charge of the polyplexes [21]. Subsequently, these modifications of the PEI-based vectors’ properties allowed for the injection of highly concentrated formulations and the reduction of deleterious interactions of the particles with blood components. On the other hand, PEGylation reduces the DNA-binding capacity of the cationic polymer, accompanied by sterical hindrance of specific interactions between polyplexes and targeted cells [22], a fact that considerably reduces in vitro transfection efficiency.

Herein, we report the detailed study of a series of frameworks (DCFs) containing PEGylated squalene [8,23], poly-(ethyleneglycol)-bis(3-aminopropyl) of different PEG units lengths, and branched low molecular weight polyethylenimine components, reversibly connected in a hyperbranched structure as efficient dynamic non-viral vectors. The interaction of DCFs with nucleic acids, as well as delivery of plasmid DNA to human cells, were investigated using gel electrophoresis, AFM and pCS2+MT-Luc DNA transfection in HeLa cells. In addition, the data on the influence of the poly-(ethyleneglycol)-bis(3-aminopropyl) length in composition of designed frameworks on the size, transfection efficiency and tolerance in human cells were analyzed and compared.

## 2. Materials and Methods

### 2.1. Materials

PEGylated squalene (SQ-PEG-NH_2_) was synthesized according to previous reports and this is briefly described in ESI [8,23]. Poly-(ethyleneglycol)-bis(3-aminopropyl) (average MW 1500 g/mol—PEG-1500; average MW 2000 g/mol—PEG-2000); average MW 3000 g/mol—PEG-3000), branched polyethylenimine (average MW 800 g/mol—bPEI800), acetonitrile (≥99.9%), sucrose (≥99.5%), tris(hydroxymethyl)aminomethane (≥99.8%), ethylenediaminetetraacetic acid anhydrous (≥99%), and acetic acid (≥99%) were purchased from Sigma–Aldrich GmbH (Steinheim, Germany). Benzene-1,3,5-tricarbaldehyde 98% (TA) was purchased from Manchester Organics Ltd (Runcorn, Cheshire, England, United Kingdom. Ethidium bromide solution 1% was purchased from AppliChem GmbH (Darmstadt, Germany). Ultrapure water was obtained with TKA-GenPure 08.2204 system. Cell experiments: Human cervical carcinoma cell line (HeLa) were purchased from CLS-Cell-Lines-Services-GmbH, Eppelheim, Germany. Alpha-MEM medium, fetal bovine serum (FBS), 1× Trypsin-Versene (EDTA) mixture, and 1% Penicillin-Streptomycin-Amphotericin B mixture (10 K/10 K/25 µg in 100 mL) were purchased from Lonza (Basel, Switzerland). Phosphate buffered saline (PBS) was purchased from Invitrogen (Paisley, Scotland, UK). Bright Glo Luciferase Assay Reagent and CellTiter 96^®^ Aqueous One Solution Cell Proliferation Assay were purchased from Promega (Madison, WI, USA). All chemicals were utilized without any further purification.

### 2.2. Experimental

#### 2.2.1. Synthesis of Non-Viral Vectors (NV) Libraries

Synthesis of NV1-NV30 was performed in three steps [8,9,10]. Step 1: in a 100 mL pear-shaped flask PEGylated squalene [8,23], (379.5 mg, 0.204 mmol, 1 equivalent) was solubilized in 20 mL acetonitrile (ACN). Next, benzene-1,3,5-tricarbaldehyde (TA) (33 mg, 0.204 mmol, 1 equivalent), solubilized in 10 mL ACN, was directly added to the PEGylated squalene solution. The obtained reaction mixture was magnetically stirred under nitrogen atmosphere at room temperature (25 °C) for 24 h. The reaction completion was monitored by recording ^1^H-NMR spectra of the reaction mixture (Appendix A). Step 2: the reaction solution from step 1 was dissolved in ACN (12.5 mg, 0.0062 mmol, 1 equivalent) and mixed with different molar ratios of H_2_N-PEG-NH_2_ (1500 Da, 2000 Da and 3000 Da according to Appendix A) in ACN (3 mL). Step 3: the ACN from solutions in Step 2 was removed in vacuo; the remained residues were solubilized in ultrapure water (2 mL) at 40 °C under mild magnetic stirring and used in further experiments with no additional purification. Next, bPEI800 (7.44 mg, 9.3 nmol, 1.5 equivalent) dissolved in 0.5 mL ultrapure water was added to each aqueous sample solution and allowed to stir at ambient temperature (25 °C) for 48 h. The obtained mixtures were kept in aqueous solution at 2–4 °C for further investigations at known concentrations (Appendix A).

#### 2.2.2. Preparation of NV/pDNA Polyplexes

Polyplexes were prepared according to following procedures: the utilized N/P ratio represented the molar ratio of the N (nitrogen) from amino groups in the vector related to the P from phosphate groups in the plasmid DNA. Plasmid DNA (pCS2+MT-Luc) (500 ng/µL) was mixed with corresponding NV solutions at different N/P ratios and incubated at room temperature for 30–60 min to generate NV/pDNA polyplexes.

### 2.3. Methods

#### 2.3.1. Nuclear Magnetic Resonance (NMR)

^1^H-NMR and ^13^C-NMR spectra were recorded on Bruker Avance III 400 instrument operated at 400 and 100 MHz, respectively, at room temperature (23 °C). Chemical shifts are reported in ppm, and using tetramethylsilane (TMS) as internal standard. Samples were prepared by solubilizing approximately 15 mg of completely dried compound in 0.6 mL deuterated solvent (CDCl_3_ or D_2_O). The obtained spectra were edited with MestReNova 6.0.2-5475 from Mestrelab Research S.L.

#### 2.3.2. Transmission Electron Microscopy (TEM)

TEM analyses were accomplished using HT7700 Hitachi Transmission Electron Microscope. DCFs aqueous solutions were diluted with ultrapure water to desired concentrations and volumes of 6 μL were deposited on 400 mesh carbon coated copper grids (purchased from TED PELLA). The grids were air dried for 24 h at ambient temperature (23 °C) in dust free conditionsand the samples were examined in high resolution mode. Size distributions of DCFs were established by measuring the diameter of 90 particles using ImageJ 1.48r, a free image processing program developed by the National Institute of Health (NIH).

#### 2.3.3. Mean Size and Zeta Potential

Mean size and zeta potential of the obtained systems were evaluated by DelsaNano C Submicron Particle Size Analyzer from Beckman. Light source: Dual 30 mW laser diodes, 658 nm. For zeta potential, the instrument used electrophoretic light scattering (ELS) to measure the zeta potential, which determined electrophoretic movement of charged particles under an applied electric field. Measurements were performed at 25 °C and the final result represented an average of six measurements. The analysis mode used the Smoluchowski equation. Each sample was dispersed in ultrapure water to obtain desired concentrations with a final volume of 2 mL. Size measurements were achieved using a 10 mm size glass cell module with the following software settings: Accumulation times—70, scattering angle—165°, correlation method—TD, attenuator 1—100%, pinhole—50 μm. Zeta measurements were recorded using a Flow Cell module with the following software settings: accumulation times—25 (five accumulations in five different points), scattering angle—15°, correlation method—TD, attenuator 1—73.2%, attenuator 2—3.84%, pinhole—50 μm. The obtained results were processed with Delsa Nano Software Version 3.73 from Beckman Coulter Inc.

#### 2.3.4. Atomic Force Microscopy (AFM)

AFM investigations were performed on Ntegra Spectra instrument (NT-MDT, Russia) operated in tapping mode under ambient conditions. Silicon cantilever tips (NSG 10) with a resonance frequency of 140–390 kHz, a force constant of 5.5–22.5 N m1, and a tip curvature radius of 10 nm were utilized. Samples were prepared by depositing 6 μL of NV/pDNA polyplexe solutions at N/P 50 on freshly cleaved mica substrates, rinsed with water to remove buffer salts, and dried in air at room temperature (23 °C). Mean diameter values and size distribution of polyplexes were established by measuring the diameter of 80 particles using the ImageJ 1.48r software application, a free image processing program developed by the National Institute of Health (NIH).

#### 2.3.5. Agarose Gel Retardation Assay

Agarose gel electrophoresis was applied to evaluate the polyplexes formation. Both the naked plasmid DNA (pCS2+MT-Luc) and the obtained polyplexes, prepared from pCS2+MT-Luc (0.5 μg) and NV1-NV30 at different N/P ratios (1, 2, 3, 4, 5, 10, 15, 20, 50, 100, 150, 200), were mixed with loading buffer (10× TAE buffer, pH 7.4) and then loaded per well in a 1% agarose gel. Electrophoresis was carried out at 90 V for 90 min in 1× TAE running buffer solution (40 mM Tris-HCl, 1% glacial acetic acid, 1 mM EDTA). The migration of free and complexed pCS2 was visualized and photographed under UV light using a MiniBIS Pro system from (DNR Bio-Imaging) after staining with ethidium bromide (15 μL of 1% ethidium bromide in 300 mL double distilled water) and incubated for 20 min in the dark at ambient temperature (~25 °C).

#### 2.3.6. Preparation of Plasmid DNA

Plasmid pCS2+MT-Luc, which encodes firefly luciferase (Harvard University, Boston) and pCS2+NLS-eGFP, which encodes enhanced green fluorescent protein (Harvard University, Boston), were propagated by molecular cloning in *Escherichia coli* DH5α, extracted and purified with an E.Z.N.A. Endo-free Plasmid Mini II kit (Omega Bio-Tek, Inc., Norcross, Georgia, USA.

#### 2.3.7. Cell Cultures

HeLa cells were cultivated in tissue culture flasks with alpha-MEM medium supplemented with 10% fetal bovine serum and 1% Penicillin-Streptomycin-Amphotericin B mixture (10 K/10 K/25 µg in 100 mL). The medium was changed for a fresh one every 3–4 days. After reaching confluence, cells were detached with 1× Trypsin-Versene (EDTA) mixture, washed with phosphate buffered saline, centrifuged at 200 × g for 3 min and subcultured into new tissue culture flasks. All cell culture experiments were conducted under the same conditions: in the humidified incubator at 37 °C and 5% CO_2_.

#### 2.3.8. In vitro Gene Transfection Study

Cells (HeLa) were seeded 24 h prior to transfection into a 96-well culture plate (from Corning) with a density of 1 × 10^4^ cells per well in 100 µL culture medium (alpha-MEM medium supplemented with 10% fetal bovine serum (FBS) and 1% Penicillin-Streptomycin-Amphotericin B mixture (10 K/10 K/25 µg in 100 mL)). For transfection, the medium in each well was replaced with 100 µL of mixture containing fresh medium and NV/pDNA polyplexes containing 500 ng pCS2+MT-Luc plasmid DNA per well, at concentrations corresponding to N/P 50 and 100. After 48 h in the humidified incubator at 37 °C and 5% CO_2_, 100 µL Bright Glo Luciferase Assay Reagent was added to each well, and the luminescence was measured within a 4 min interval. Six biological replicates were performed for each sample.

#### 2.3.9. In vitro Mitochondrial Reductase Activity Assay (MTS) Cytotoxicity Study

The in vitro MTS cytotoxicity study was achieved using the CellTiter 96^®^ Aqueous One Solution Cell Proliferation Assay. HeLa cells were seeded into a 96-well culture plate at a density of 1 × 10^4^ cells per well in 100 µL culture medium (alpha-MEM medium supplemented with 10% fetal bovine serum (FBS) and 1% Penicillin-Streptomycin-Amphotericin B mixture (10 K/10 K/25 µg in 100 mL)). After 24 h, the medium in each well was replaced with 100 µL mixture containing fresh medium and NV/pDNA polyplexes containing 500 ng pCS2+NLS-eGFP plasmid DNA per well at N/P ratios of 50 and 100. Six biological replicates were performed for each sample. After 44 h in the humidified incubator at 37 °C and 5% CO_2_, 20 μL of CellTiter 96^®^ Aqueous One Solution reagent was added to each well, and the plates were incubated for another 4 h prior to results reading. Absorbance at 490 nm was recorded with a plate reader (EnSight, from PerkinElmer, Singapore). Cell viability was calculated and expressed as percentage relative to viability of untreated cells.

#### 2.3.10. Statistical Analysis

Statistical analyses were performed using GraphPad Prism 6.04 for Windows (GraphPad Software, La Jolla California, CA, USA). A Student’s t-test was applied for comparison between NV/pDNA polyplexes and references samples, with 6 replicates for each group included. Results are presented as means ± standard deviation (S.D.) and the difference was considered statistically significant when values of calculated probability (*p*) were lower than 0.05.

## 3. Results and Discussion

### 3.1. Synthesis of Library NV1–NV30

A library of DCF non-viral vectors (NV1–NV30, Appendix A) was designed and prepared following previously described procedure [8,9]. The synthesis involved cross-linking the network components and DNA-binding sites via the amino-carbonyl/imine reversible chemistry to a core molecule. Thus, (i) the hydrophobic component SQ-PEG-NH_2_, known for its self-assembly ability in aqueous solutions [8,23,24,25,26]; (ii) H_2_N-PEG-NH_2_ (Mn: ~1500, 2000 and 3000 Da) segments, known to favor solubility in water and to reduce the immunogenicity of the systems; and (iii) bPEI (Mn: ~800 Da) bPEI800 as cationic sites, able to bind DNA, were connected to the TA core in different ratios. Treatment of TA with different equivalents of SQ-PEG-NH_2_ and subsequent addition of H_2_N-PEG-NH_2_ in ACN resulted in the formation of a mixture of linear and cross-linked structures [6,8,9]. Using 1.5 equivalent of PEI in the last step of the synthesis proved to be the optimal ratio related to the best transfection efficiency of DCF [9]. Thus, in preparation of the NV1–NV30 library, the ratio of TA:SQ-PEG-NH_2_:bPEI800 remained constant (1:1:1.5). The ratio of H_2_N-PEG-NH_2_ was gradually changed from 0.1 to 1 equivalent and the MW of H_2_N-PEG-NH_2_ from 1500 to 2000 and 3000 Da, resulting in the flowing vectors: NV1–NV10 (containing 0.1–1 equivalent of H_2_N-PEG-NH_2_ of 1500 Da), NV11–NV20 (containing 0.1–1 equivalent of H_2_N-PEG-NH_2_ of 2000 Da) and NV21–NV30 (containing 0.1–1 equivalent of H_2_N-PEG-NH_2_ of 3000 Da) (see Appendix A). The mechanism of DCFs formation occurred in three steps (Figure 1): First, the reaction between TA and SQ-PEG-NH_2_ in the molar ratio of 1:1, performed in ACN for 24 h, followed by the addition of H_2_N-PEG-NH_2_ in established ratio and the last one, after evaporation of the solvent, the residue was suspended in water followed by the addition of 1.5 equivalent of bPEI800.

### 3.2. Morphological and Coloidal Stability of DCFs

Since the prepared NV libraries are prone to formation of hydrophob/hydrophil self-assembly in water [8,11,23], our interest was focused on comparing the contribution of PEG’s size from 1500 to 3000 Da, with the use of H_2_N-PEG-NH_2_ on formed particles. Thus, detailed TEM and DLS analysis for the NV10, NV20 and NV30 representatives was performed. Analyzing TEM images, the formation of spherical entities for all the investigated DCFs could be observed (Figure 2). Interestingly, changing the molecular weight of H_2_N-PEG-NH_2_ from 1500 Da (NV10, Figure 2a) to 2000 Da (NV20, Figure 2b), and further to 3000 Da (NV30, Figure 2c), led to a decrease in particle size, showing the clear influence of the PEG unit length on the overall framework size. A similar effect of PEG-dependent vector size decrease in PEI-based non-viral vectors was observed by Kumar et al. [14]. In their work, the increase of charged PEG units number within the vector has led to the increase of interactions with PEI moieties, leading to a clear decrease of the sizes of PEI-PEG vector nanoparticles.

The TEM data were well supported by the DLS analysis (Figure 3). In the case of NV10, at different analyzed concentration, a strong decay of hydrodynamic size of the particles with the increase of the concentration was observed (Figure 3a), presenting a behavior similar to micelles [27]. When analyzing NV20 and NV30 DCF solutions, we could observed that the determined concentration/hydrodynamic size decay was less evident due to the fact that larger PEG units may participate in stabilization of the formed particles by limiting the aggregation processes due to the long PEG chains repulsions [28,29,30].

To elucidate the effect of the length of the PEG side chain on the surface coverage, we measured the zeta potential for the vectors NV10, NV 20 and NV30 at different concentrations. Zeta potential (Figure 3b) suggested that NV10 has a low colloidal stability and its strength grows with the increase of concentration. These data corroborate the DLS analysis, since the aggregation processes of NV10 results in large structures. For NV20 and NV30, however, only a slight decrease of zeta potential was observed. This low decay indicates that the larger PEG units within the NV20 and NV30 frameworks not only stabilize the framework, but also participate in the charge shielding effect [29]. This nonaggregating effect could be also observed if zeta potential data were correlated with TEM results, NV20 and NV30 assemblies had shown smaller sizes when compared to the micellar-like behavior of the NV10 framework.

Overall, replacing of H_2_N-PEG-NH_2_ of 1500 Da with corresponding 2000 and 3000 Da in NV20 and NV30, respectively, led to higher stability of the particles at lower concentrations (4 × 10^−5^ M–8 × 10^−5^ M), with a further increase in concentration leading to agglomeration in the case of shorter PEG units [27,31].

### 3.3. Plasmid DNA Binding Abilities

In the next step, the ability of NV10, NV20, NV30 (Figure 4a–c), as well as NV1, NV4, NV7, NV11, NV14, NV17, NV21, NV24, NV27 (Appendix A), to bind pCS2+MT-Luc was evaluated by agarose gel electrophoresis and their electrophoretic mobility was examined at various N/P ratios in comparison to bPEI800 Da (Figure 4d). Thus, NV10 fully bind the pDNA beginning with N/P ratios of 10, forming NV10/pDNA polyplex (Figure 4a), while NV20 at N/P ratios of 20, forming NV20/pDNA the polyplex (Figure 4b), and those of NV30 at N/P ratios of 50 forming the NV30/pDNA polyplex (Figure 4c).

It can be observed that increasing of the H_2_N-PEG-NH_2_ length from 1500 Da to 3000 Da in composition of NVs translated to a weakening of the pDNA binding ability (Appendix A). The observed dependency could be attributed to a degree of hindrance of bPEI800 units by the length of PEG units used in the framework’s formation [28].

The interaction of NVs with pDNA, resulting in the formation of corresponding polyplexes, was additionally analyzed by atomic force microscopy (AFM). AFM images (Figure 5) were recorded for NV10/pDNA, NV20/pDNA, NV30/pDNA in order to observe how the length of PEG influenced the shape and size of the studied polyplexes.

In order to provide a long-drawn circulation in the vascular system, the polyplex nanoparticles must have the dimension between 10 and 200 nm. Otherwise, the particles below 10 nm are rapidly cleared through the kidney; meanwhile, the entities above 200 nm are susceptible to clearance by the reticuloendothelial system (RES) [32,33,34]. In this paradigm, the dimensions of the newly formed polyplexes represent an important characteristic, being absolutely essential to be checked [35,36]. In the case of NV10/pDNA polyplexes at N/P ratio 50, the AFM analysis revealed well defined spherically-shaped particles (Figure 5a) with the size distributions of ~130 nm (Figure 5a′). In this context, the observed sizes of the NV10/pDNA polyplexes appeared to be suitable for cellular uptake by cells. On the other hand, NV20/pDNA and NV30/pDNA polyplexes (Figure 5b,c) showed the formation of larger particles with sizes between 460 and 560 nm, respectively (Figure 5b′,c′), indicating a clear influence of the length of PEG unit over the mechanism of polyplex formation. These preliminary data on the sizes of the NV20 and NV30 polyplexes may indicate difficulties of their cellular uptake.

### 3.4. Cytotoxicity and Transfection Efficiency

To confirm the preliminary conclusions that resulted from AFM data on the sizes of NV10, NV20 and NV30 polyplexes, in vitro transfection efficiency and cytotoxicity studies were performed (Figure 6 and Figure 7).

NV1-NV30/pDNA polyplexes were formed at N/P ratio 50 and 100 and tested on HeLa cellular line using Bright-Glo(TM) Luciferase Assay, using bPEI800 as a reference, being a part of the composition of the synthesized vectors. Transfection tests were performed in three series: (*i*) NV1/pDNA-NV10/pDNA, containing H_2_N-PEG-NH_2_ of 1500 Da in their compositions, starting with the ratio 0.1 equiv. for NV1 to 1 equiv. for NV10; (*ii*) NV11/pDNA-NV20/pDNA and NV21/pDNA-NV30/pDNA, containing H_2_N-PEG-NH_2_ of 2000 and 3000, from 0.1 to 1 equivalent, respectively (Figure 6). While analyzing the NV1 / pDNA-NV10 / pDNA series (Figure 6a), a gradual increase in the transfection efficiency with increasing the ratio of H2N-PEG-NH2 from 0.1 to 1 equivalent was clearly observed, reaching the best transfection value for NV9/pDNA polyplex. For the NV11/pDNA-NV20/pDNA and NV21/pDNA-NV30/pDNA series, the optimal transfection results were evaluated for NV20/pDNA and NV29/pDNA; however, no strong dependence on the used ratio of H_2_N-PEG-NH_2_ was observed (Figure 6b,c). For these analyzed samples, the transfection efficiency appeared to be higher at N/P ratio 100 compared to N/P ratio 50.

In order to assess the toxicity NV/pDNA polyplexes, a mitochondrial reductase activity assay (MTS) was used [37,38]. All the NVs polyplexes were tested in vitro using MTS cytotoxicity to HeLa cells and obtained results are graphical represented in Figure 6a′,b′,c′; the cell viability was calculated and expressed as percent relative to the viability of untreated cells (100% viability). When analysing polyplexes containing H_2_N-PEG-NH_2_ 1500 Da (Figure 6a′), the relative cell viability was insignificantly influenced by the molar ratio of the PEG polymer in the vector, at N/P ratio of 50 the cell viability was around 80%, whereas at N/P ratio of 100, the cell viability dropped to ~60 %. Interestingly, it has been observed that by replacing H_2_N-PEG-NH_2_ 1500 Da with H_2_N-PEG-NH_2_ 2000 and 3000 Da the relative cell viability increased at N/P 50 and 100 (Figure 6b′,c′), suggesting that higher molecular weights of PEG can increase the biocompatibility [39] or the cell uptake was much lower due to the size of the formed polyplex particles. Similar to H_2_N-PEG-NH_2_ 1500 Da, the H_2_N-PEG-NH_2_ of 2000 or 3000 Da polyplexes’ relative cell viability was insignificantly influenced by the molar ratio of PEG within the vector structure.

Despite the fact that NV9/pDNA and NV29/pDNA showed the best transfection efficiency, NV10/pDNA, NV20/pDNA and NV30/pDNA were chosen as the representative polyplexes, having similar composition and being more susceptible for comparison with previously obtained DCF [8,9,10]. These frameworks were subjected to transfection efficiency and cytotoxicity independent tests for comparison, in order to highlight the differences induced by the length of H_2_N-PEG-NH_2_. (Figure 7). Thus, evaluating the transfection efficiency shown in Figure 7a, the advantage of NV10 in terms of transfection efficiency at both investigated N/P ratios can be observed. Additionally, for N/P ratio 100, transfection efficiency appeared to be higher when compare to N/P 50. As for cytotoxicity, NV20/pDNA and NV30/pDNA revealed higher biocompatibility compared to NV10/pDNA (Figure 7b), suggesting that higher molecular weights of PEG increase the framework’s biocompatibility. However, low toxicity and low transfection efficiency of NV20/pDNA and NV30/pDNA can be caused by lower uptake of polyplexes due to the formation of large size particles, as supported by AFM images (Figure 5b,c). Although PEGylation provided shielding and stealth properties to PEI/DNA polyplexes, in some cases they are also known to reduce the non-specific ionic interactions between the complex and target cells, thus decreasing overall TE [40,41].

## 4. Conclusions

A library of modular dynamic vectors with tuneable composition was prepared using earlier described protocols, containing hydrophobic component SQ-PEG-NH_2_, poly-(ethylene-glycol)-bis (3-amino-propyl)-terminated of various molecular weight (Mn 1500, 2000 and 3000 Da) NH_2_-PEG-NH_2_ segments, and low molecular weight branched polyethyleneimine (Mn 800 Da) bPEI800 linked to TA core. The library was designed to gradually increase both the H_2_N-PEG-NH_2_ ratio and the molecular weight of H_2_N-PEG-NH_2_ (from 1500 Da to 3000 Da) into the vectors compositions. It was observed from TEM and DLS data that the increase in molecular weight of H_2_N-PEG-NH_2_ led to the formation of smaller size particles due to the sterical interactions of PEG units with the framework’s components. It was also observed that increasing the length of H_2_N-PEG-NH_2_ from 1500 Da to 3000 Da in the composition of NVs led to a weaker binding ability of pDNA with a larger polyplex nanoparticle formation, due to the PEG shielding effect over the bPEI800 units. While analyzing NV1/pDNA-NV30/pDNA polyplexes for the transfection efficiency on HeLa cell line, we observed that by increasing the ratio of H_2_N-PEG-NH_2_ from 0.1 to 1 equivalent the transfection efficiency was significantly improved. Generally, for all analyzed samples the transfection efficiency was higher at N/P ratio 100 when compared to N/P ratio 50. The molecular weight of H_2_N-PEG-NH_2_ presented an overall significant influence on both the transfection efficiency and cell viability. Thus, the use of H_2_N-PEG-NH_2_ with relatively high molecular weight inside the dynamic framework increased the biocompatibility of the vector, but decreased the in vitro transfection efficiency.

## Figures and Tables

**Figure 1 polymers-11-01313-f001:**
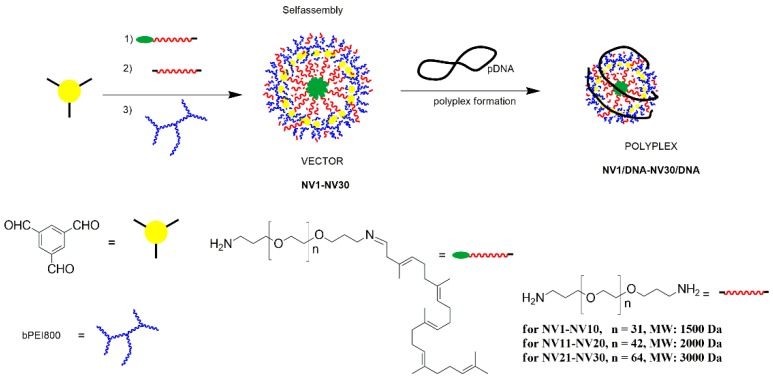
Schematic representation for the formation of vectors NV1–NV30 and polyplexes.

**Figure 2 polymers-11-01313-f002:**
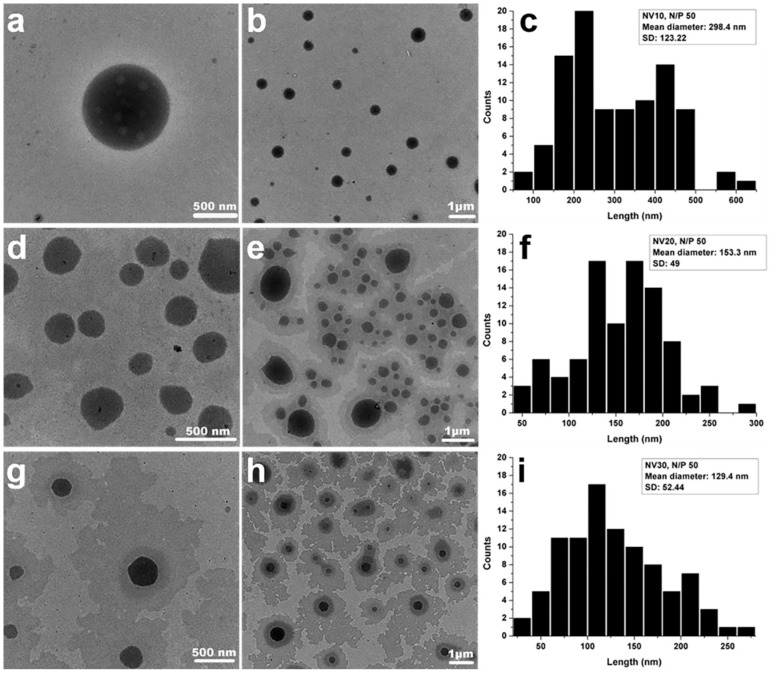
TEM micrographs of NV10 (**a**,**b**), NV20 (**d**,**e**), NV30 (**g**,**h**); and corresponding average size distributions in water for: NV10 (**c**), NV20 (**f**) and NV30 (**i**). For the size distribution path 90 particles were measured.

**Figure 3 polymers-11-01313-f003:**
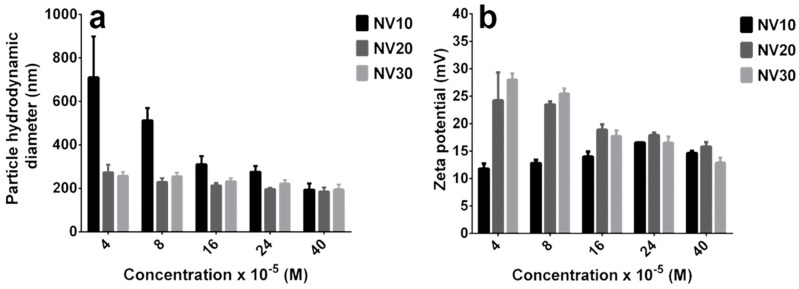
Particle hydrodynamic diameter of NV10, NV20 and NV30 at various concentrations (**a**). Zeta potential of NV10, NV20 and NV30 at various concentrations (**b**). The utilized concentrations were calculated reporting to bPEI800 weight from dynamic frameworks aqueous solutions.

**Figure 4 polymers-11-01313-f004:**
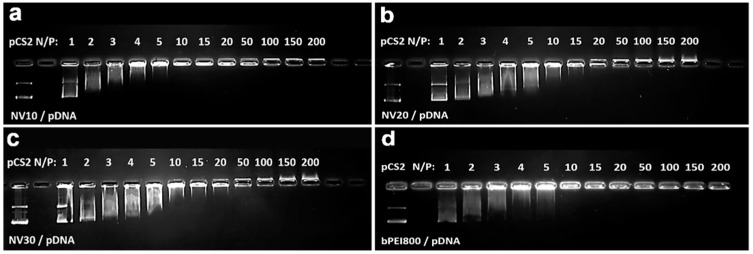
Electrophoretic mobility of polyplexes formed by complexation of pDNA and NV10, NV20, NV30 at various N/P ratios. (**a**) NV10/pDNA, (**b**) NV20/pDNA, (**c**) NV30/pDNA and (**d**) bPEI800/pDNA.

**Figure 5 polymers-11-01313-f005:**
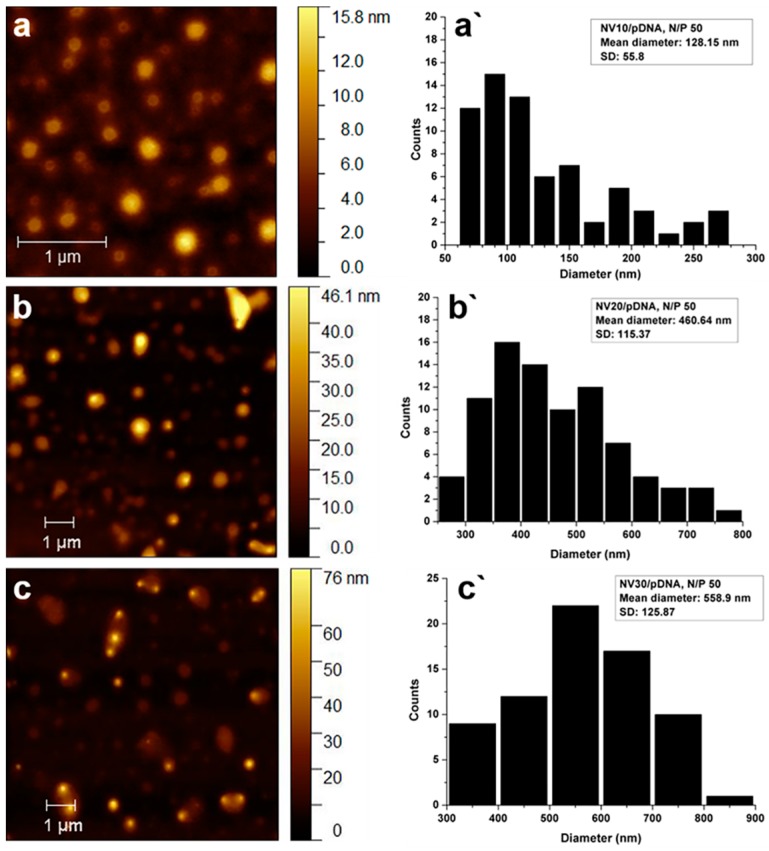
AFM imaging for (**a**) NV10/pDNA polyplex, (**b**) NV20/pDNA polyplex and (**c**) NV30/pDNA polyplex. Size distribution for (**a′**) NV10/pDNA polyplex, (**b′**) NV20/pDNA polyplex and (**c′**) NV30/pDNA polyplex at N/P ratio of 50; 80 particles were measured to obtain diameter distribution path.

**Figure 6 polymers-11-01313-f006:**
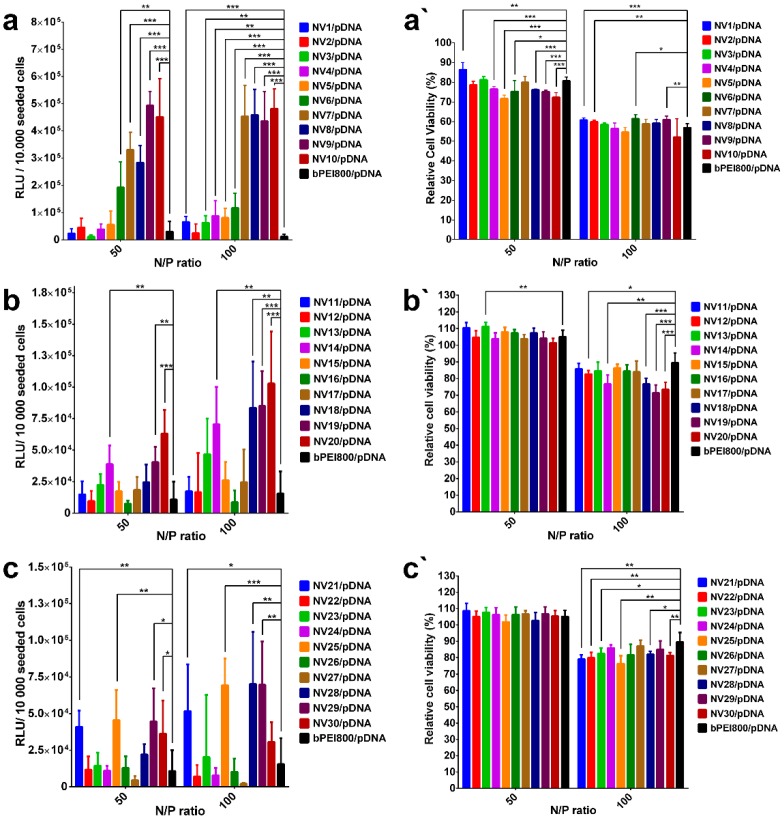
In vitro transfection efficiency of NV1-NV30 on HeLa cells (results given in relative light units (RLU) per 10000 cells): (**a**) Polyplexes containing PEG 1500 Da, (**b**) polyplexes containing PEG 2000 Da, (**c**) polyplexes containing PEG 3000 Da. In vitro MTS cytotoxicity tests on HeLa cells: (**a′**) Polyplexes containing PEG 1500 Da, (**b′**) polyplexes containing PEG 2000 Da, (**c′**) polyplexes containing PEG 3000 Da. The results are presented as a mean value ± the standard deviation (S.D.); n = 6–8. * *p* < 0.05, ** *p* < 0.01, and *** *p* < 0.001 by Student’s t-test.

**Figure 7 polymers-11-01313-f007:**
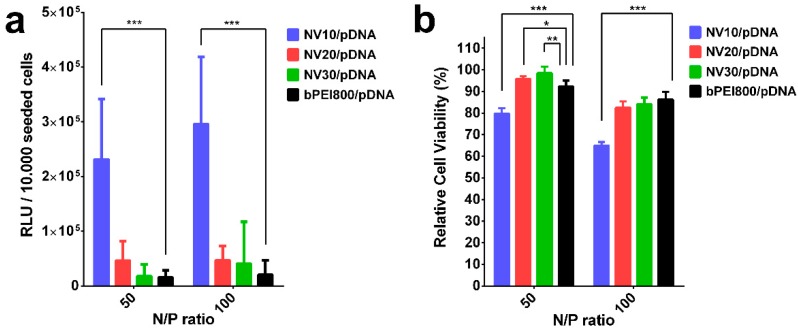
In vitro biological assessments of NV10, NV20, NV30 on HeLa cells: (**a**) Transfection efficiency, (**b**) cytotoxicity. The results are presented as a mean value ± the standard deviation (S.D.); n = 6–8. * *p* < 0.05, ** *p* < 0.01, and *** *p* < 0.001 by Student’s t-test.

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
