# Peer review of "Tunable Composition of Dynamic Non-Viral Vectors over the DNA Polyplex Formation and Nucleic Acid Transfection"

_polymers, 2019, doi:10.3390/polym11081313_

Round 1
Reviewer 1 Report
Gen,h)” have to be not only described but also discussed and justified. The Authors ha veto
General comment
The submitted manuscript reports on the preparation and characterisation of dynamic combinatorial frameworks containing PEGylated squalene, poly-(ethyleneglycol)-bis(3-aminopropyl) of different lengths, and branched low molecular weight polyethylenimine components.
The topic of the present work is worthy of investigation, and well fits the aim and scope of Polymers.
The Authors failed to compare their data with literature results. Thus, they should improve the Results and Discussion section, adding recent and appropriate literature reports.
Moreover, a deep English grammar and language revision is strongly recommended.
More details and specific remarks and suggestions are reported below point by point.
Abstract
- All the acronyms have to be written in the extended form the first time they are used, such as TEM, DLS, AFM and so on.
Keywords
Among the keywords, some about the application have to be added.
1. Introduction
- The Introduction section is well organised and well-conceived. However, the Authors have to highlight the originality and added value of their work with respect to the literature and, particularly, with respect to their own previous papers about the same topic.
- More details about PEG have to be reported, considering its properties and main applications and citing related recent literature references, such as “Injectable Silk Fibroin-Hydrogels Functionalized with Microspheres as Adult Stem Cells-Carrier Systems, International Journal of Biological Macromolecules 108(2018): 960-971”.
- The following conclusion “The obtained frameworks were capable to form distinct 70 supramolecular amphiphilic architectures, driven by self-assembly properties of squalene 71 derivatives, thus cumulating a specific number of PEI units per carrier” sounds out of place and more suitable for the Abstract and/or Conclusions sections.
- It is suggested to report a brief list of the used characterisations at the end of the Introduction section.
-
2. Materials and Methods
2.1. Materials
Even if synthesised according previous reports, the synthesis procedure of PEGylated Squalene (SQ-PEG-NH2) has to briefly reported.
2.3.3. Mean size and zeta potential
The following phrase “Have been evaluated using an DelsaNano C Submicron Particle Size Analyzer from Beckman Coulter equipped with dual 30 mW laser diodes emitting at 658 nm.” has to be correctly rewritten.
2.3.2. Transmission electron microscopy (TEM)
- The Authors have to specify how many particles they considered for the measurements with ImageJ in order to provide an average value and a standard deviation.
2.3.4. Atomic force microscopy (AFM)
- The Authors have to specify how many particles they considered for the measurements with ImageJ in order to provide an average value and a standard deviation.
2.3.5. Agarose gel retardation assay
The following phrase “Was applied to electrophoretically evaluate the formation of the polyplexes” has to be correctly rewritten.
3. Results and Discussion
3.2. Morphological and coloidal stability of DCFs
As a general comment, the Authors failed in discussing and comparing the acquired data with the literature. They should not only describe but also discuss their results.
- The following considerations “Interestingly, changing the molecular weight of H2N-PEG-NH2 from 1500 Da (NV10, Figure 2a) to 2000 Da (NV20, Figure 2b) and further to 3000 Da (NV30, Figure 2c), has led to the decrease in particle size, showing the clear influence of the PEG unit length on the overall framework size. It was also observed that in the case of NV20 and NV30, containing the longer PEG, a partial agglomeration of frameworks took place (Figure 2e,h)” have to be not only described but also discussed and justified. The Authors ha veto corroborate them with suitable literature references.
- Similarly the following statement “Zeta potential, on the other hand (Figure 3b), suggested that NV10 have a low colloidal stability and its strength growths with the increase of concentration. For NV20 and NV30 however, only a slight decrease of zeta potential was observed” have to be better justified and compared with literature.
3.3. Plasmid DNA binding abilities
- The following considerations “The observed dependency could be attributed to a degree of hindrance of bPEI800 units by the length of PEG units used in the frameworks formation” and “On the other hand, NV20/pDNA and NV30/pDNA polyplexes (Figure 5b, c) showed the formation of larger particles with sizes between 460 and 560 nm, respectively (Figure 5b’, c’), indicating a clear influence of the length of PEG unit over the mechanism of polyplex formation.”need literature references to be corroborated.
4. Conclusions
The Conclusion section is too long and should be made more concise, in order to highlight the main results.
Author Response
Dear Reviewer,
We acknowledge for your time dedicated to analysis of our work. We have revised the manuscript according to your suggestions and we are grateful that due to your observations we could considerably improve the scientific quality of our manuscript.
In the following, the detailed responses to the comments are presented. All modifications suggested by the Reviewers are made to the main manuscript using track changes.
General comments
The submitted manuscript reports on the preparation and characterisation of dynamic combinatorial frameworks containing PEGylated squalene, poly-(ethyleneglycol)-bis(3-aminopropyl) of different lengths, and branched low molecular weight polyethylenimine components.
The topic of the present work is worthy of investigation, and well fits the aim and scope of Polymers.
The Authors failed to compare their data with literature results. Thus, they should improve the Results and Discussion section, adding recent and appropriate literature reports.
Moreover, a deep English grammar and language revision is strongly recommended.
More details and specific remarks and suggestions are reported below point by point.
Abstract
- All the acronyms have to be written in the extended form the first time they are used, such as TEM, DLS, AFM and so on.
Answer: Thank you, it is corrected and we corrected in main manuscript.
Keywords
Among the keywords, some about the application have to be added.
Answer: Thank you, two additional keywords were added.
1. Introduction
The Introduction section is well organised and well-conceived. However, the Authors have to highlight the originality and added value of their work with respect to the literature and, particularly, with respect to their own previous papers about the same topic. More details about PEG have to be reported, considering its properties and main applications and citing related recent literature references, such as “Injectable Silk Fibroin-Hydrogels Functionalized with Microspheres as Adult Stem Cells-Carrier Systems, International Journal of Biological Macromolecules 108(2018): 960-971”.
Answer: The introduction section was modified according to the Reviewers suggestions and completed with corresponding bibliographic reports.
- The following conclusion “The obtained frameworks were capable to form distinct 70 supramolecular amphiphilic architectures, driven by self-assembly properties of squalene 71 derivatives, thus cumulating a specific number of PEI units per carrier” sounds out of place and more suitable for the Abstract and/or Conclusions sections.
Answer: Thank you, this part has been removed.
- It is suggested to report a brief list of the used characterisations at the end of the Introduction section.
Answer: The brief description of the characterization techniques has been introduced.
2. Materials and Methods
2.1. Materials
Even if synthesised according previous reports, the synthesis procedure of PEGylated Squalene (SQ-PEG-NH2) has to briefly reported.
Answer: The synthesis of SQ-PEG-NH2 was added to the ESI in the revised version of the submission.
2.3.3. Mean size and zeta potential
The following phrase “Have been evaluated using an DelsaNano C Submicron Particle Size Analyzer from Beckman Coulter equipped with dual 30 mW laser diodes emitting at 658 nm.” has to be correctly rewritten.
Answer: Thank you, it is corrected we did correction in the main manuscript.
2.3.2. Transmission electron microscopy (TEM)
- The Authors have to specify how many particles they considered for the measurements with ImageJ in order to provide an average value and a standard deviation.
Answer: Thank you, the information was added.
2.3.4. Atomic force microscopy (AFM)
- The Authors have to specify how many particles they considered for the measurements with ImageJ in order to provide an average value and a standard deviation.
Answer: Thank you, the information was added.
2.3.5. Agarose gel retardation assay
The following phrase “Was applied to electrophoretically evaluate the formation of the polyplexes” has to be correctly rewritten.
Answer: Thank you, it is corrected, we did corrections.
3. Results and Discussion
3.2. Morphological and coloidal stability of DCFs
As a general comment, the Authors failed in discussing and comparing the acquired data with the literature. They should not only describe but also discuss their results.
- The following considerations “Interestingly, changing the molecular weight of H2N-PEG-NH2 from 1500 Da (NV10, Figure 2a) to 2000 Da (NV20, Figure 2b) and further to 3000 Da (NV30, Figure 2c), has led to the decrease in particle size, showing the clear influence of the PEG unit length on the overall framework size. It was also observed that in the case of NV20 and NV30, containing the longer PEG, a partial agglomeration of frameworks took place (Figure 2e,h)” have to be not only described but also discussed and justified. The Authors have to corroborate them with suitable literature references.
Answer: The obtained data on the NV10, NV20 and NV30 were discussed and the correlation with other experiments and literature was added to the corresponding discussion part.
- Similarly the following statement “Zeta potential, on the other hand (Figure 3b), suggested that NV10 have a low colloidal stability and its strength growths with the increase of concentration. For NV20 and NV30 however, only a slight decrease of zeta potential was observed” have to be better justified and compared with literature.
Answer: We have introduced additional discussions based on the obtained results. However, we could not compare the obtained results with similar literature reports since, to the best of our knowledge, there are no literature report describing the mechanism of self-assembly and interaction of dynamic combinatorial frameworks systems containing PEG units.
3.3. Plasmid DNA binding abilities
- The following considerations “The observed dependency could be attributed to a degree of hindrance of bPEI800 units by the length of PEG units used in the frameworks formation” and “On the other hand, NV20/pDNA and NV30/pDNA polyplexes (Figure 5b, c) showed the formation of larger particles with sizes between 460 and 560 nm, respectively (Figure 5b’, c’), indicating a clear influence of the length of PEG unit over the mechanism of polyplex formation.”need literature references to be corroborated.
Answer: Literature report reflecting the corresponding statements has been introduced to the manuscript.
4. Conclusions
The Conclusion section is too long and should be made more concise, in order to highlight the main results.
Answer: Thank you, it has been modified.
Reviewer 2 Report
This work by Clima et al. describes the systematic evaluation of the effect of varying amounts of H2N-PEG-NH2 of three different molecular weights (1500, 2000 and 3000 Da) in the structure and activity of dynamic combinatorial frameworks (DCFs) for DNA delivery. Thirty different formulations were prepared and characterized by TEM, DLS, AFM and agarose gel retardation assay. The transfection capacity on HeLa cells was finally compared employing pCS2+MT-Luc DNA as a reporter gene.
This manuscript builds on previous work on similar DCF systems by the authors. Even though this is not the first time they describe such structures, this work does fill a gap in the understanding of these systems, and provides novel insights that can be seized to optimize their performance.
In general, the work is well presented, the methods are well described and appropriate for the goals planned, and the main conclusions are supported by the presented data. However, there are a few issues with some of its contents that could be improved in a revised version of the manuscript.
Major comments:
1. Even though DLS results were presented for the “empty” NV10, NV20 and NV30 formulations, the size evaluation of the polyplexes appears to have been based only on AFM data (Page 8, Figure 5). I believe that including the DLS evaluation of these samples would improve the quality of this measurements (given the statistical nature of DLS compared to imaging-based evaluation). The Z potential measurements of the polyplexes could also reveal the interaction with the nucleic acid and would be interesting to be presented.
2. On page 9, line 310-311, the authors say that “For the NV11/pDNA-NV20/pDNA and NV21/pDNA-NV30/pDNA series the optimal transfection results were evaluated for NV20 and NV30…”. Upon evaluation of Figure 6 (on the same page), part of this statement appears not to be true, since the transfection efficiency of NV29 appears to be larger than that of NV30. Could the authors please clarify this point?. This same issue arises again on page 10, line 330-331, when it is said that “NV10/pDNA, NV20/pDNA and NV30/pDNA were the representative polyplexes, being most effective vectors from each series, were subjected to transfection efficiency and cytotoxicity tests…”. In this case, I believe that it is still reasonable to compare these three formulations (given their similarities in composition), but if the data presented on Figure 6 are correct, the reasoning about them being the “most effective vectors from each series” should be removed.
3. It is unclear to me whether the data presented in Figure 7 correspond to an independent experiment from the data in Figure 6, or they are just a comparison among representative candidates from parts a, b and c from Figure 6. Could you please clarify this point?.
4. Have the authors considered that the differences in toxicity and transfection efficiency of the samples (commented on page 10, lines 320-323, and again on lines 332-337) could be both derived from the larger size of NV20 and NV30 compared to NV10? (as seen in Figure 5). Since larger particle sizes could result in lower uptake, both decreased transfection and toxicity could be expected in NV20 and NV30 from this perspective. I believe this should be discussed in the paper with the reasons in favor or against this hypothesis that the authors may have.
Minor comments:
1. It is unclear to me whether the data presented really supports that difference in the formation mechanism take place, or just that differences in polyplex structure arise from the compositional variations. I would ask the authors to consider this point and maybe modify the title accordingly.
2. Given that multiple comparisons were made, for example in Figure 6, have authors considered performing some statistical correction for these multiple comparisons? (Related to the statistical analysis method section, on page 5).
3. Some typographical / grammatical errors can be found (although these do not prevent the correct understanding of the manuscript), and I would suggest a revision of the text to improve this parameter. As an example, on page 10, line 332, “the transfection efficiency sown in” instead of “shown in”.
Author Response
Dear Reviewer,
We acknowledge for your time dedicated to analysis of our work. We have revised the manuscript according to your suggestions and we are grateful that due to your observations we could considerably improve the scientific quality of our manuscript.
In the following, the detailed responses to the comments are presented. All modifications suggested by the Reviewers are made to the main manuscript using track changes.
Comments and Suggestions for Authors
This work by Clima et al. describes the systematic evaluation of the effect of varying amounts of H2N-PEG-NH2 of three different molecular weights (1500, 2000 and 3000 Da) in the structure and activity of dynamic combinatorial frameworks (DCFs) for DNA delivery. Thirty different formulations were prepared and characterized by TEM, DLS, AFM and agarose gel retardation assay. The transfection capacity on HeLa cells was finally compared employing pCS2+MT-Luc DNA as a reporter gene.
This manuscript builds on previous work on similar DCF systems by the authors. Even though this is not the first time they describe such structures, this work does fill a gap in the understanding of these systems, and provides novel insights that can be seized to optimize their performance.
In general, the work is well presented, the methods are well described and appropriate for the goals planned, and the main conclusions are supported by the presented data. However, there are a few issues with some of its contents that could be improved in a revised version of the manuscript.
Major comments:
1. Even though DLS results were presented for the “empty” NV10, NV20 and NV30 formulations, the size evaluation of the polyplexes appears to have been based only on AFM data (Page 8, Figure 5). I believe that including the DLS evaluation of these samples would improve the quality of this measurements (given the statistical nature of DLS compared to imaging-based evaluation). The Z potential measurements of the polyplexes could also reveal the interaction with the nucleic acid and would be interesting to be presented.
Answer: We agree with the observation that size and zeta potential of polyplexes are important information that could improve the manuscript. However, our main concern and difficulty regarding these measurements are related to high amount and high costs of extracted plasmid DNA that is needed for the measurements (milligram quantities) using available instrumentation. We have tried to use a more accessible option using commercial model DNA (Deoxyribonucleic acid sodium salt from salmon testes which is a double stranded DNA containing ~ 2000 bp) to perform required measurements after its interaction with NV10, NV20 and NV30 frameworks. From the performed experiment (table below), it was observed that the zeta potential value decreases from 6.72 to 2.20 mV while the vectors interact with model DNA and with increasing of PEG chain in polyplex composition. This data are in good correlation with literature (10.1002/jbm.a.31343), explaining that the increasing of PEG chain in polyplex will produce the lowering of the surface charge of the resulted complex owing to PEG shielding effects.
|
Zeta potential, mV |
||
|
Vector |
Nanoparticle |
DNA loading complex (N/P=50) |
|
NV10 |
14.64 |
6.72 |
|
NV20 |
15.83 |
4.36 |
|
NV30 |
12.91 |
2.20 |
We decided to not introduce and discuss these results since experiments were performed using salmon DNA instead of utilized plasmid DNA in the manuscript experiments, thus the obtained data are not completely applicable.
2. On page 9, line 310-311, the authors say that “For the NV11/pDNA-NV20/pDNA and NV21/pDNA-NV30/pDNA series the optimal transfection results were evaluated for NV20 and NV30…”. Upon evaluation of Figure 6 (on the same page), part of this statement appears not to be true, since the transfection efficiency of NV29 appears to be larger than that of NV30. Could the authors please clarify this point?. This same issue arises again on page 10, line 330-331, when it is said that “NV10/pDNA, NV20/pDNA and NV30/pDNA were the representative polyplexes, being most effective vectors from each series, were subjected to transfection efficiency and cytotoxicity tests…”. In this case, I believe that it is still reasonable to compare these three formulations (given their similarities in composition), but if the data presented on Figure 6 are correct, the reasoning about them being the “most effective vectors from each series” should be removed.
Answer: Thank you, it is corrected and completed.
3.It is unclear to me whether the data presented in Figure 7 correspond to an independent experiment from the data in Figure 6, or they are just a comparison among representative candidates from parts a, b and c from Figure 6. Could you please clarify this point?
Answer: The data from figure 7 correspond to an independent experiment with the aim to underline the differences between the chosen polyplexes. The corresponding comments were also highlighted in the manuscript as well.
4. Have the authors considered that the differences in toxicity and transfection efficiency of the samples (commented on page 10, lines 320-323, and again on lines 332-337) could be both derived from the larger size of NV20 and NV30 compared to NV10? (as seen in Figure 5). Since larger particle sizes could result in lower uptake, both decreased transfection and toxicity could be expected in NV20 and NV30 from this perspective. I believe this should be discussed in the paper with the reasons in favor or against this hypothesis that the authors may have.
Answer: We indeed agree with mentioned statement and introduced this argument into the manuscript.
Minor comments:
1. It is unclear to me whether the data presented really supports that difference in the formation mechanism take place, or just that differences in polyplex structure arise from the compositional variations. I would ask the authors to consider this point and maybe modify the title accordingly.
Answer: Thank you for this observation. We agree on changing the title of the manuscript to “Tunable composition of dynamic non-viral vectors over the DNA polyplex formation and nucleic acid transfection” as it suits better to the manuscript content.
2. Given that multiple comparisons were made, for example in Figure 6, have authors considered performing some statistical correction for these multiple comparisons? (Related to the statistical analysis method section, on page 5).
Answer: We are grateful for this suggestion. We have not considered the statistical correction in this project. We were more interested in the elucidation of the PEG length influence over the dynamic system structure and their properties. Since indeed, there are multiple comparisons within the study of dynamic systems, we will take this part into consideration in the future work.
3. Some typographical / grammatical errors can be found (although these do not prevent the correct understanding of the manuscript), and I would suggest a revision of the text to improve this parameter. As an example, on page 10, line 332, “the transfection efficiency sown in” instead of “shown in”.
Answer: Thank you, spelling mistakes were checked and corrected.
Reviewer 3 Report
Authors focused on the preparation of PEGylated non-viral vectors and their further investigations. Manuscript has been prepared properly. It is based on the extensive Introduction section explaining the background of undertaken research subject. Next, suitable methodology has been applied and obtained results have been analyzed correctly. In general, proposed paper seems very interesting but some revisions are recommended because some provided information needs to be described in more detail.
Firstly, some statements included in Introduction need to be developed to increase the scientific value of the manuscript, e.g. “An important aspect that has intensively been studied in recent years was the surface charge shielding effect of the vectors by hydrophilic polymers, in particular by polyethylene glycols (PEGs)”- this issue needs to be briefly described by Authors. The same applies to the synergistic effect of low molecular weight polyethylenimine and poly(ethylene glycol) components of a non-viral vector structure. The mentioned synergy phenomenon should also be briefly discussed.
Next, procedure of the synthesis of PEGylated Squalene (SQ-PEG-NH2) needs to be briefly described. The same applies to the MTS cytotoxicity assay. Its main principle has to be presented by Authors.
Moreover, some conclusions drawn on the basis of the conducted research are too general and need to be more widely analyzed. Authors mentioned e.g. that “Interestingly, changing the molecular weight of H2N-PEG-NH2 (…) has led to the decrease in particle size, showing the clear influence of the PEG unit length on the overall framework size”- the mentioned dependence of the molecular weight and the particle size should be described in more detail. Additionally, Authors stated that “(…) higher molecular weights of PEG increased the biocompatibility (…)” and this sentence also should be developed.
What is more, quality of figures needs to be improved.
Author Response
Dear Reviewer,
We acknowledge for your time dedicated to analysis of our work. We have revised the manuscript according to your suggestions and we are grateful that due to your observations we could considerably improve the scientific quality of our manuscript.
In the following, the detailed responses to the comments are presented. All modifications suggested by the Reviewers are made to the main manuscript using track changes.
Comments and Suggestions for Authors
Authors focused on the preparation of PEGylated non-viral vectors and their further investigations. Manuscript has been prepared properly. It is based on the extensive Introduction section explaining the background of undertaken research subject. Next, suitable methodology has been applied and obtained results have been analyzed correctly. In general, proposed paper seems very interesting but some revisions are recommended because some provided information needs to be described in more detail.
Firstly, some statements included in Introduction need to be developed to increase the scientific value of the manuscript, e.g. “An important aspect that has intensively been studied in recent years was the surface charge shielding effect of the vectors by hydrophilic polymers, in particular by polyethylene glycols (PEGs)”- this issue needs to be briefly described by Authors. The same applies to the synergistic effect of low molecular weight polyethylenimine and poly(ethylene glycol) components of a non-viral vector structure. The mentioned synergy phenomenon should also be briefly discussed.
Answer: Thank you, the introduction section was improved.
Next, procedure of the synthesis of PEGylated Squalene (SQ-PEG-NH2) needs to be briefly described. The same applies to the MTS cytotoxicity assay. Its main principle has to be presented by Authors.
Answer: Thank you, the synthesis of PEGylated Squalene (SQ-PEG-NH2) was added to ESI. MTS protocol was described in section 2.3.9. In vitro (MTS) cytotoxicity study.
Moreover, some conclusions drawn on the basis of the conducted research are too general and need to be more widely analyzed. Authors mentioned e.g. that “Interestingly, changing the molecular weight of H2N-PEG-NH2 (…) has led to the decrease in particle size, showing the clear influence of the PEG unit length on the overall framework size”- the mentioned dependence of the molecular weight and the particle size should be described in more detail. Additionally, Authors stated that “(…) higher molecular weights of PEG increased the biocompatibility (…)” and this sentence also should be developed.
Answer: Thank you, mentioned sections were improved. Additional discussions were added to the manuscript.
What is more, quality of figures needs to be improved.
Answer: Thank you, quality of figures were improved to 300 dpi.
Round 2
Reviewer 1 Report
The Authors have followed all the Reviewers’ suggestions and now the paper looks very improved and can be accepted in the present version.
Reviewer 2 Report
I believe that the authors have improved the quality of the manuscript, which is now suitable for publication at Polymers.
Reviewer 3 Report
Comparing current revised version of the paper with its previous version it may be observed that the paper has been improved by Authors. All comments of the Reviewer have been discussed. Some statements from the Introduction section indicated by the Reviewer have been developed, e.g. issue concerning the surface charge shielding effect of the vectors by PEG units as well as the cooperative effect between PEI and PEG moieties. Authors have also described in more detail synthesis of PEGylated Squalene that has been presented in Supplementary Material. Some too general sentences formulated in the analysis of obtained results have also been changed. Therefore, paper in its current, corrected version may be accepted for publication in the Journal.